# Association between insulin administration method and six-month neurological outcome in survivors of out-of-hospital cardiac arrest who underwent targeted temperature management

**Dong Hun Lee[1], Byung Kook Lee[1,2]\*, Yong Soo Cho[1], Yong Hun Jung[1,2], Hyoung Youn Lee[2,3], Kyung Woon Jeung[1,2], Chun Song Youn[4], Soo Hyun Kim[4], on behalf of Korean Hypothermia Network Investigators[¶]**

1 Department of Emergency Medicine, Chonnam National University Hospital, Gwangju, Republic of Korea, 2 Department of Emergency Medicine, Chonnam National University Medical School, Gwangju, Republic of Korea, 3 Trauma center, Chonnam National University Hospital, Gwangju, Republic of Korea, 4 Department of Emergency Medicine, College of Medicine, The Catholic University of Korea, Seoul, Republic of Korea

¶ Membership of Korean Hypothermia Network is provided in the Acknowledgments.
\* bbukkuk@hanmail.net

## Abstract

We investigated the association of insulin administration method with the achievement of mean glucose ≤ 180 mg/dL and neurological outcomes in out-of-hospital cardiac arrest (OHCA) survivors who had hyperglycemia after the return of spontaneous circulation. From a multicenter prospective registry, we extracted the data of adult OHCA survivors who underwent targeted temperature management (TTM) between 2015 and 2018. Blood glucose levels every 4 h after initiating TTM were obtained for 72 h. We divided insulin administration methods into three categories: subcutaneous (SQI), intravenous bolus (IBI), and continuous intravenous (CII). We calculated the mean glucose and standard deviation (SD) of glucose. The primary outcome was the achievement of mean glucose ≤ 180 mg/dL. The secondary outcomes were the 6-month neurological outcome based on the Cerebral Performance Category (CPC) scale (good, CPC 1–2; poor, CPC 3–5), mean glucose, and SD of glucose. Of the 549 patients, 296 (53.9%) achieved mean glucose ≤ 180 mg/dL, and 438 (79.8%) had poor neurological outcomes, 134 (24.4%), 132 (24.0), and 283 (51.5%) were in the SQI, IBI, and CII groups, respectively. The SQI (adjusted odds ratio [aOR], 0.848; 95% confidence intervals [CIs], 0.493–1.461) and IBI (aOR, 0.673; 95% CIs, 0.415–1.091) groups were not associated with mean glucose ≤ 180 mg/dL and the SQI (aOR, 0.660; 95% CIs, 0.335–1.301) and IBI (aOR, 1.757; 95% CIs, 0.867–3.560) groups were not associated with poor neurological outcomes compared to the CII group. The CII (168 mg/dL [147–202]) group had the lowest mean glucose than the SQI (181 mg/dL [156–218]) and IBI (184 mg/dL [162–216]) groups. The CII (45.0[33.9–63.5]) group had a lower SD of glucose than the IBI (50.8 [39.1–72.0]) group. The insulin administration method was not associated with achieving mean glucose ≤ 180 mg/dL and 6-month neurological outcomes.

**Data Availability Statement:** All relevant data are within the manuscript and its Supporting Information files.

**Funding:** This study was supported by a grant from National Research Foundation of Korea (NRF-2021R1A2C1005800). BK Lee received this fund. The funders had no role in the study design, data collection and analysis, publication decision, or manuscript preparation. There was no additional external funding received for this study.

**Competing interests:** The authors have declared that no competing interests exist.

## Introduction

Cardiac arrest leads to various metabolic derangements due to ischemic-reperfusion injury even after the return of spontaneous circulation (ROSC). Hyperglycemia is one of the common abnormalities following cardiac arrest [1]. A well-known scientific observation is the association between hyperglycemia and poor neurological outcomes or increased mortality in cardiac arrest survivors [2–5]. Therefore, the target glucose level is generally recommended as < 180 mg/dL, this parameter is applied to critically ill patients [6], as in comatose cardiac arrest survivors, as protection against potential neurological injury, even if the ultimate target glycemic range has not been elucidated [7, 8].

In a historic comparison study, although the target glucose range of the continuous intravenous insulin (CII) method changed from conventional range (150 to 200 mg/dL) to intensive range (100 to 150 mg/dL) in patients with coronary artery bypass graft, the CII method has been proven to reduce mortality compared to the subcutaneous insulin (SQI) method [9]. Therefore, intravenous insulin infusion is the preferred route for administering insulin in critically ill patients [10]. A randomized controlled trial has revealed that conventional glucose control (< 180 mg/dL) was associated with lower mortality than intensive glucose control (81 to 108 mg/dL) in critically ill patients [6]. However, a randomized controlled trial that compared strict and conventional glucose control in comatose cardiac arrest survivors failed to find the optimal target glucose range [11]. Furthermore, although insulin is administrated subcutaneously or intravenously (bolus or continuous), the clinical difference according to the insulin administration in cardiac arrest survivors has not been investigated.

To address this question, we hypothesized that the insulin administration method would be related to blood glucose levels and, thus, to neurological outcomes. To examine this hypothesis, we used a multicenter registry of out-of-hospital cardiac arrest (OHCA) who underwent targeted temperature management (TTM) with blood glucose recordings for 72 h after the initiation of TTM.

## Materials and methods

### Study design and population

The Korean Hypothermia Network Prospective Registry (KORHN-PRO) has been gathering data on comatose adult (age ≥ 18 years) OHCA survivors who underwent TTM at 20 participating hospitals since October 2015 (KORHN-PRO; NCT02827422) [12]. The KORHN-PRO collects data on blood glucose after ROSC and every 4 h from the initiation of TTM to 72 h. We performed a retrospective analysis of the KORHN-PRO data between October 2015 and December 2018. The ethics and institutional review board of all participating hospitals approved KORHN-PRO. Written informed consent was obtained from all patients or patients' proxies per national requirements and the principle of the Declaration of Helsinki [13]. Independent researchers assessed the neurological outcomes at 1 and 6 months after ROSC and recorded these as the Cerebral Performance Category (CPC) scale [14].

We included adult OHCA survivors who had hyperglycemia (> 180 mg/dL) within 24 h following ROSC. We excluded patients who had: insufficient data on glucose (glucose measurement less than six times during the 72 h after the initiation of TTM); no insulin for glucose control; no data on insulin treatment; no hyperglycemia within 24 h following ROSC; no data on 6-month neurological outcomes; died within 24 h after ROSC.

### TTM and glucose control

After ROSC, the target temperature range of 33–36˚C was achieved as soon as possible. Sedatives were administered, and if needed, neuromuscular blockade to control shivering. Patients

were rewarmed after completing the maintenance phase at 0.2–0.5˚C/h. Sedatives and neuro-muscular blockade were discontinued as the patient achieved normothermia. Blood glucose level was monitored and controlled to avoid hyperglycemia or hypoglycemia throughout post-cardiac arrest care following ROSC. Hyperglycemia was managed with SQI, intravenous bolus insulin (IBI), or CII according to each hospital protocol and the attending physician (S1 File). Moderate (< 70 mg/dL) or severe hypoglycemia (< 40 mg/dL) was managed with glucose-containing solution.

## Data collection

We extracted the following data from the KORHN-PRO: age; sex; body mass index (BMI); pre-existing illness; a witness of collapse; bystander cardiopulmonary resuscitation (CPR); first monitored rhythm (shockable or non-shockable); etiology of cardiac arrest (cardiac or non-cardiac); time from collapse to ROSC; epinephrine dose; serum lactate level after ROSC, $PaO_2$ and $PaCO_2$ after ROSC; glucose levels after ROSC and every 4 h during the 72 h from the initiation of TTM; sequential organ failure assessment (SOFA) within the first day following ROSC [15]; target temperature (33–34˚C or 35–36˚C); insulin administration method (SQI, IBI, or CII); CPC 6-months after ROSC.

The primary outcome was the achievement of mean blood glucose ≤ 180 mg/dL. The secondary outcomes were neurological outcomes assessed using CPC 6 months after ROSC, maximum glucose, mean glucose, a standard deviation (SD) of glucose, minimum glucose, moderate hypoglycemia, and severe hypoglycemia. The neurological outcomes were defined as good (CPC 1 or 2) or poor (CPC 3–5).

## Statistical analysis

We report continuous variables as median with interquartile ranges because all continuous variables had a non-normal distribution and categorical variables as the frequency with per-centile. We used chi-square or Fisher's exact test to compare categorical variables, as appropriate. We used the Mann–Whitney $U$ test to compare continuous variables between two groups and the Kruskal–Wallis test to compare continuous variables among three groups. We performed posthoc analysis using a pair-wise Mann–Whitney $U$ test with Bonferroni correction. We conducted logistic regression analyses to investigate the association between the insulin administration method and the achievement of mean blood glucose ≤ 180 mg/dL and the association between insulin administration methods and neurological outcomes. We selected the variables with a $p$-value < 0.05 in comparisons among insulin administration methods as covariates for the association between insulin administration methods and mean blood glucose ≤ 180 mg/dL. Additionally, we selected the covariates after performing the multivari-able logistic regression analysis with the variables with a p-value < 0.2 in comparisons between groups of mean glucose ≤ 180 mg/dL and > 180 mg/dL. We also selected the covariates for the association between insulin administration methods and neurological outcomes through the multivariable logistic regression analysis with variables with a $p$-value < 0.2 in compari-sons between neurological outcome groups. We selected the variables with a $p$-value < 0.05 in the multivariate logistic regression analyses as final covariates. We performed the Hosmer–Lemeshow test to test the goodness-of-fit of the logistic model. We also performed the multi-variate logistic regression analysis to examine the association between glucose variables and neurological outcomes after adjusting covariates. We report the logistic regression analysis results as an adjusted odds ratio (aOR) with 95% confidence interval (CIs). We used IBM SPSS Statistics 26.0 for Windows (IBM Corp., Armonk, NY). A two-sided $p$-value < 0.05 was used to indicate statistical significance.

## Results

### Study population

Of 1,373 OHCA survivors who were recorded in the registry, 289 patients had no hyperglycemia during 24 h after ROSC; 251 patients were either not administered insulin or lacked insulin data; 152 patients had insufficient blood glucose measurement (< 6 times) data during the 72 h after ROSC; 97 patients died or transferred within 24 h; 35 patients had no available data on 6-month CPC. Finally, 549 patients were included in the study (Fig 1).

### Characteristics according to the insulin administration method

Table 1 shows the baseline and clinical characteristics stratified by the insulin administration method. The SQI, IBI, and CII groups comprised 134 (24.4%), 132 (24.0%), and 283 (51.5%) patients, respectively. Pre-existing diabetes mellitus and renal disease, a witness of collapse, and etiology of cardiac arrest were different among the three groups. Time from collapse to ROSC and $PaCO_2$ were different among the three groups. However, subgroup analyses found no difference in time from collapse to ROSC and $PaCO_2$ between paired two groups. Pre-TTM shock and target temperature were different among the three groups.

Table 2 shows the glucose variables according to the insulin administration method. Achievement of mean glucose ≤ 180 mg/dL, mean glucose, SD of glucose, and minimum glucose differed among the three groups. The CII group (168 mg/dL [147–202]) had lower mean glucose levels than the SQI (181 mg/dL [156–218]) and IBI (184 mg/dL [162–216]) groups. The IBI group (50.8 [39.1–72.0]) had higher SDs of glucose than the SQI (47.0 [33.2–61.1]) and CII (45.0 [33.9–63.5]) groups. The CII group (98 mg/dL [82–112]) had lower minimum glucose levels than the SQI group (106 mg/dL [91–122]). However, the IBI group had similar minimum glucose levels as the SQI and CII groups. Severe hypoglycemia was different among the three groups.

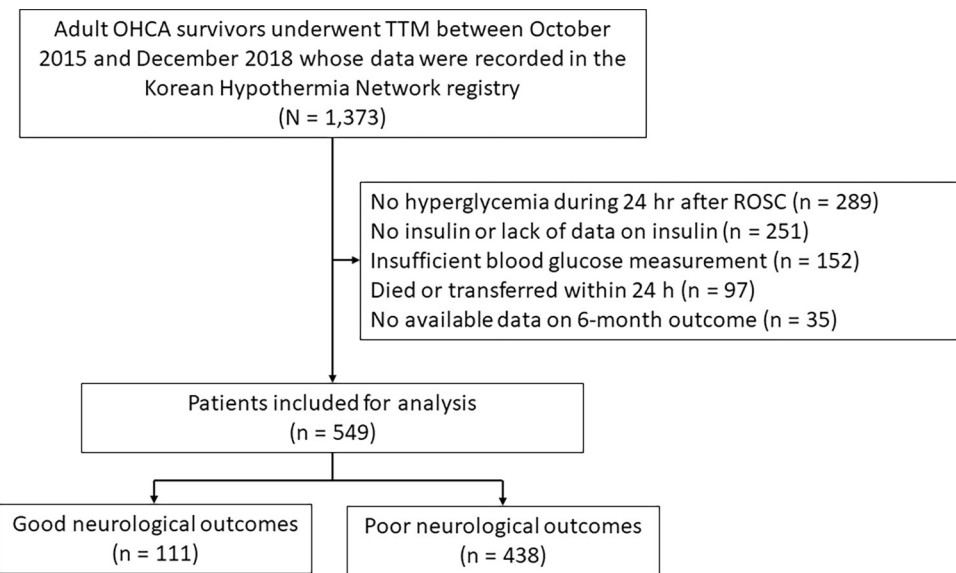

**Fig 1. Flow chart describing the patient selection.** OHCA, out-of-hospital cardiac arrest; TTM, targeted temperature management; ROSC, return of spontaneous circulation.

**Table 1. Baseline and clinical characteristics stratified by the insulin administration method.**

| Variables | SQI (n = 134) | IBI (n = 132) | CII (n = 283) | p |
|---|---|---|---|---|
| Age, years | 60.0 (51.8–71.0) | 60.0 (49.0–70.0) | 61.0 (51.0–71.0) | 0.769 |
| Male sex | 82 (61.2) | 97 (73.5) | 190 (67.1) | 0.102 |
| BMI, kg/m$^2$ | 23.0 (20.8–25.4) | 23.8 (20.2–25.9) | 23.5 (21.1–26.0) | 0.652 |
| Pre-existing illness | | | | |
| CAD | 15 (11.2) | 21 (15.9) | 33 (11.7) | 0.410 |
| Heart failure | 7 (5.2) | 9 (6.8) | 9 (3.2) | 0.232 |
| Hypertension | 56 (41.8) | 53 (40.2) | 133 (47.0) | 0.352 |
| Diabetes mellitus | 38 (28.4) | 59 (44.7) | 98 (34.6) | 0.019 |
| Stroke or TIA | 14 (10.4) | 12 (9.1) | 15 (5.3) | 0.126 |
| Pulmonary disease | 7 (5.2) | 13 (9.8) | 21 (7.4) | 0.357 |
| Renal disease | 7 (5.2) | 19 (14.4) | 19 (6.7) | 0.010 |
| Liver cirrhosis | 0 (0.0) | 1 (0.8) | 4 (1.4) | 0.357 |
| Witnessed | 67 (50.0) | 91 (68.9) | 204 (72.1) | <0.001 |
| Bystander CPR | 78 (58.2) | 86 (65.2) | 174 (62.4) | 0.508 |
| Shockable rhythm | 42 (31.3) | 36 (27.3) | 85 (30.0) | 0.755 |
| Cardiac etiology | 72 (53.7) | 83 (62.9) | 190 (67.1) | 0.030 |
| Time to ROSC, min | 37.0 (21.0–49.0)[a] | 30.0 (19.0–43.0)[a] | 32.0 (18.0–45.0)[a] | 0.039 |
| Epinephrine dose, mg | 2 (1–5), 130 | 2 (1–4), 129 | 2 (1–4), 266 | 0.968 |
| Serum lactate, mg/dL | 10.3 (7.0–12.9) | 10.7 (7.0–14.0) | 9.9 (6.3–13.0) | 0.283 |
| Glucose affter ROSC, mg/dL | 292 (213–356) | 288 (228–372) | 285 (231–355) | 0.855 |
| PaO$_2$, mmHg | 102.1 (70.0–193.0) | 107.0 (74.4–180.8) | 109.3 (73.0–196.0) | 0.407 |
| PaCO$_2$, mmHg | 45.1 (33.9–61.8)[a] | 53.0 (39.0–75.9)[a] | 50.0 (37.4–72.7)[a] | 0.040 |
| Pre-TTM shock | 94 (70.1) | 89 (67.4) | 148 (52.3) | <0.001 |
| SOFA | 12 (10–13) | 12 (10–13) | 11 (9–13) | 0.141 |
| Target temperature | | | | |
| 33˚C–34˚C | 63 (47.0) | 114 (86.4) | 270 (95.4) | <0.001 |
| 35˚C–36˚C | 71 (53.0) | 18 (13.6) | 13 (4.6) | |

SQI, subcutaneous insulin; IBI, intravenous bolus insulin; CII, continuous intravenous insulin; BMI, body mass index; CAD, coronary artery disease; TIA, transient ischemic attack; CPR, cardiopulmonary resuscitation; ROSC, return of spontaneous circulation; TTM, targeted temperature management; SOFA, sequential organ failure assessment

[a]Post-hoc analysis by pair-wise Mann–Whitney U test with Bonferroni correction showed no difference.

**Table 2. Glucose characteristics during 72 h after cardiac arrest according to the insulin administration method.**

| Characteristics | SQI (n = 134) | IBI (n = 132) | CII (n = 283) | p |
|---|---|---|---|---|
| MG ≤ 180 mg/dL | 65 (48.5) | 61 (46.2) | 170 (60.1) | 0.011 |
| Mean, mg/dL | 181 (156–218)[a] | 184 (162–216)[a] | 168 (147–202)[b] | 0.001 |
| Maximum, mg/dL | 338 (269–415) | 357 (285–417) | 325 (268–389) | 0.080 |
| SD | 47.0 (33.2–61.1)[a] | 50.8 (39.1–72.0)[b] | 45.0 (33.9–63.5)[a] | 0.012 |
| Minimum, mg/dL | 106 (91–122)[a] | 101 (82–114)[a,b] | 98 (82–112)[b] | 0.008 |
| Moderate hypoglycemia | 11 (8.2) | 13 (9.8) | 34 (12.0) | 0.475 |
| Severe hypoglycemia | 0 (0.0) | 8 (6.1) | 9 (3.2) | 0.017 |

SQI, subcutaneous insulin; IBI, intravenous bolus insulin; CII, continuous intravenous insulin; MG, mean glucose; SD, standard deviation

[a,b]Post-hoc analysis by pair-wise Mann–Whitney U test with Bonferroni correction. The same letter means no difference.

**Table 3. Demographic and clinical characteristics stratified by mean glucose of 180 mg/dL.**

| Variables | Total (n = 549) | MG ≤ 180 mg/dL (n = 296) | MG > 180 mg/dL (n = 253) | p |
|---|---|---|---|---|
| Age, years | 61.0 (50.0–71.0) | 59.0 (49.0–71.0) | 62.0 (53.5–71.0) | 0.093 |
| Male sex | 369 (67.2) | 199 (67.2) | 170 (67.2) | 1.000 |
| Body mass index, kg/m$^2$ | 23.5 (20.9–25.8) | 23.0 (20.7–25.4) | 23.9 (21.3–26.1) | 0.025 |
| Pre-existing illness | | | | |
| Coronary artery disease | 69 (12.6) | 31 (10.5) | 38 (15.0) | 0.141 |
| Heart failure | 25 (4.6) | 12 (4.1) | 13 (5.1) | 0.688 |
| Hypertension | 242 (44.1) | 112 (37.8) | 130 (51.4) | 0.002 |
| Diabetes mellitus | 195 (35.5) | 63 (21.3) | 132 (52.2) | <0.001 |
| Stroke or TIA | 41 (7.5) | 18 (6.1) | 23 (9.1) | 0.240 |
| Pulmonary disease | 41 (7.5) | 28 (9.5) | 13 (5.1) | 0.079 |
| Renal disease | 45 (8.2) | 25 (8.4) | 20 (7.9) | 0.941 |
| Liver cirrhosis | 5 (0.9) | 2 (0.7) | 3 (1.2) | 0.860 |
| Witnessed | 362 (65.9) | 208 (70.3) | 154 (60.9) | 0.026 |
| Bystander CPR | 338 (61.6) | 185 (62.5) | 153 (60.5) | 0.690 |
| Shockable rhythm | 163 (29.7) | 98 (33.1) | 65 (25.7) | 0.072 |
| Cardiac etiology | 345 (62.8) | 184 (62.2) | 161 (63.6) | 0.789 |
| Time to ROSC, min | 32.0 (19.0–45.0) | 32.0 (19.0–45.0) | 33.0 (19.0–46.0) | 0.365 |
| Epinephrine dose, mg | 2 (1–4) | 2 (0–4) | 2 (1–5) | 0.017 |
| Serum lactate, mg/dL | 10.2 (6.7–13.3) | 9.3 (6.0–12.6) | 11.3 (8.0–14.0) | <0.001 |
| Glucose after ROSC, mg/dL | 287 (228–360) | 263 (215–323) | 323 (242–409) | <0.001 |
| PaO$_2$, mmHg | 106.8 (73.0–191.0) | 117.5 (76.0–221.0) | 100.0 (71.0–160.0) | 0.001 |
| PaCO$_2$, mmHg | 49.6 (37.0–71.4) | 48.3 (37.1–69.5) | 54.0 (38.8–75.0) | 0.343 |
| Pre-TTM shock | 331 (60.3) | 155 (52.4) | 176 (69.6) | <0.001 |
| SOFA | 12 (9–13) | 11 (9–13) | 12 (10–14) | <0.001 |
| Target temperature | | | | 0.003 |
| 33˚C–34˚C | 447 (81.4) | 255 (86.1) | 192 (75.9) | |
| 35˚C–36˚C | 102 (18.6) | 41 (13.9) | 61 (24.1) | |

MG, mean glucose; TIA, transient ischemic attack; CPR, cardiopulmonary resuscitation; ROSC, return of spontaneous circulation; TTM, targeted temperature management; SOFA, sequential organ failure assessment

## Association between insulin administration method and mean glucose under 180 mg/dL

Table 3 shows the demographic and clinical characteristics stratified by mean glucose of 180 mg/dL. A total of 296 (53.9%) achieved mean glucose ≤ 180 mg/dL. Patients who had mean glucose ≤ 180 mg/dL had lower BMI (23.0 kg/m$^2$ [20.7–25.4]) vs. 23.9 kg/m$^2$ [21.3–26.1]) than those had mean glucose > 180 mg/dL. Patients with mean glucose ≤ 180 mg/dL had fewer pre-existing illnesses (hypertension and diabetes mellitus). Patient who had mean glucose ≤ 180 mg/dL had fewer witnesses of collapse (208/296 vs. 154/253), lower epinephrine dose (2 mg [0–4] vs. 2 mg [1–5]), lower serum lactate (9.3 mg/dL [6.0–12.6] vs. 11.3 mg/dL [8.0–14.0]), higher PaO$_2$ (117.5 mmHg [76.0–221.0] vs. 100.0 mmHg [71.0–160.0]), lower pre-TTM shock (155/296 vs. 176/253), lower SOFA scores (11 [9–13] vs. 12 [10–14]), and a higher proportion of target temperature of 33˚C–34˚C (255/296 vs. 192/253).

Multivariable logistic regression analysis found hypertension (aOR, 0.609; 95% CIs, 0.399–0.930), diabetes mellitus (aOR, 0.273; 95% CIs, 0.175–0.425), witnessed (aOR, 1.992; 95% CIs, 1.289–3.078), PaO$_2$ (aOR, 1.002; 95% CIs, 1.000–1.004), pre-TTM shock (aOR, 0.541; 95% CIs,

**Table 4. The association between insulin administration method and mean glucose $\leq$ 180 mg/dL.**

| Variables | Adjusted odds ratio (95% confidence interval) | p |
|---|---|---|
| Hypertension | 0.686 (0.458–1.027) | 0.067 |
| Diabetes mellitus | 0.274 (0.180–0.417) | < 0.001 |
| Renal disease | 1.559 (0.742–3.277) | 0.241 |
| Witnessed | 1.752 (1.156–2.656) | 0.008 |
| Cardiac etiology | 0.891 (0.575–1.381) | 0.607 |
| Time to ROSC | 0.999 (0.989–1.009) | 0.820 |
| Glucose after ROSC, mg/dL | 0.995 (0.993–0.997) | <0.001 |
| PaO$_2$, mmHg | 1.003 (1.001–1.004) | |
| PaCO$_2$, mmHg | 1.003 (0.996–1.011) | 0.387 |
| Pre-TTM shock | 0.580 (0.389–0.864) | 0.007 |
| Target temperature | | |
| 33˚C–34˚C | Reference | |
| 35˚C–36˚C | 1.819 (1.094–3.025) | 0.021 |
| Insulin administration method | | |
| CII | Reference | |
| SQI | 0.848 (0.493–1.461) | 0.553 |
| IBI | 0.673 (0.415–1.091) | 0.108 |

ROSC, return of spontaneous circulation; TTM, targeted temperature management; CII, continuous intravenous insulin; SQI, subcutaneous insulin; IBI, intravenous bolus insulin

0.355–0.826), target temperature of 35–36˚C (aOR, 0.500; 95% CIs, 0.297–0.842), glucose after ROSC (aOR, 0.995; 95% CIs, 0.993–0.997) as covariates. The insulin administration method was not associated with the achievement of mean glucose $\leq$ 180 mg/dL after adjusting covariates (Table 4).

## Association between insulin administration method and neurological outcomes

Table 5 shows the demographic and clinical characteristics according to the neurological outcomes. A total of 438 (79.8%) patients had poor neurological outcomes. Patients in the poor neurological outcome group were older (62.0 years [51.8–72.0] vs. 58.0 years [48.0–67.0]) than those in the good neurological outcome group, although BMI and pre-existing illness were not different between neurological outcome groups. Those in the poor neurological outcome group had fewer witnesses of collapse (268/438 vs. 94/111), a lower proportion of bystander CPR (256/438 vs. 82/111), a lower proportion of shockable rhythm (83/438 vs. 80/111), fewer cardiac etiology (242/438 vs. 103/111), longer time from collapse to ROSC (36.0 min [23.0–47.0] vs. 18.0 min [12.0–27.0]), and received higher epinephrine dose (2 mg [1–4] vs. 0 mg [0–2]) than those in the good neurological outcome group. Those in the poor neurological outcome group had higher serum lactate levels after ROSC (10.9 mg/dL [7.5–13.7] vs. 7.5 mg/dL [4.1–11.2]), higher PaCO$_2$ (53.7 mmHg [39.0–75.1] vs. 39.0 mmHg [32.2–52.4]), more pre-TTM shock (282/438 vs. 49/111), and higher SOFA scores (12 [10–13] vs. 10 [8–12]).

Multivariate logistic regression analysis found that age (aOR, 1.031; 95% CIs, 1.008–1.054), shockable rhythm (aOR, 0.183; 95% CIs, 0.096–0.349), cardiac etiology (aOR, 0.196; 95% CIs, 0.076–0.506), time from collapse to ROSC (aOR, 1.074; 95% CIs, 1.048–1.102), PaO$_2$ (aOR, 1.003; 95% CIs, 1.000–1.006), PaCO$_2$ (aOR, 1.015; 95% CIs, 1.001–1.029), and pre-TTM shock (aOR, 1.855; 95% CIs, 1.037–3.319) were associated with poor neurological outcome. The

**Table 5. Demographic and clinical characteristics stratified by neurological outcomes.**

| Variables | Good (n = 111) | Poor (n = 438) | p |
|---|---|---|---|
| Age, years | 58.0 (48.0–67.0) | 62.0 (51.8–72.0) | 0.006 |
| Male sex | 80 (72.1) | 289 (66.0) | 0.222 |
| Body mass index, kg/m$^2$ | 23.6 (21.5–25.3) | 23.4 (20.8–26.0) | 0.718 |
| Pre-existing illness | | | |
| Coronary artery disease | 20 (18.0) | 49 (11.2) | 0.052 |
| Heart failure | 5 (4.5) | 20 (4.6) | 0.978 |
| Hypertension | 47 (42.3) | 195 (44.5) | 0.680 |
| Diabetes mellitus | 34 (30.6) | 161 (36.8) | 0.228 |
| Stroke or TIA | 9 (8.1) | 32 (7.3) | 0.774 |
| Pulmonary disease | 6 (5.4) | 35 (8.0) | 0.355 |
| Renal disease | 5 (4.5) | 40 (9.1) | 0.112 |
| Liver cirrhosis | 1 (0.9) | 4 (0.9) | 1.000 |
| Witnessed | 94 (84.7) | 268 (61.2) | <0.001 |
| Bystander CPR | 82 (73.9) | 256 (58.4) | 0.003 |
| Shockable rhythm | 80 (72.1) | 83 (18.9) | <0.001 |
| Cardiac etiology | 103 (92.8) | 242 (55.3) | <0.001 |
| Time to ROSC, min | 18.0 (12.0–27.0) | 36.0 (23.0–47.0) | <0.001 |
| Epinephrine dose, mg | 0 (0–2) | 2 (1–4) | <0.001 |
| Serum lactate, mg/dL | 7.5 (4.1–11.2) | 10.9 (7.5–13.7) | <0.001 |
| Glucose after ROSC, mg/dL | 273 (214–334) | 291 (229–368) | 0.068 |
| PaO$_2$, mmHg | 94.0 (72.0–147.0) | 111.0 (74.1–196.2) | 0.028 |
| PaCO$_2$, mmHg | 39.0 (32.2–52.4) | 53.7 (39.0–75.1) | <0.001 |
| Pre-TTM shock | 49 (44.1) | 282 (64.4) | <0.001 |
| SOFA | 10 (8–12) | 12 (10–13) | <0.001 |
| Target temperature | | | 0.865 |
| 33°C–34°C | 91 (82.0) | 356 (81.3) | |
| 35°C–36°C | 20 (18.0) | 82 (18.7) | |

TIA, transient ischemic attack; CPR, cardiopulmonary resuscitation; ROSC, return of spontaneous circulation; TTM, targeted temperature management; SOFA, sequential organ failure assessment

insulin administration method was not associated with the 6-month neurological outcomes after adjusting covariates (Table 6).

## Association between glucose variables and neurological outcomes

Table 7 shows the glucose variables stratified by neurological outcome groups. The poor neurological outcome group had higher maximum glucose (343 mg/dL [284–410] vs. 308 mg/dL [250–366]), mean glucose (179 mg/dL [154–216] vs. 164 mg/dL [145–189]), SD (48.2 [35.9–67.0] vs. 41.1 [29.9–52.9]), and more frequent moderate hypoglycemia (52/438 vs. 6/111) than the good neurological outcome group (Table 7).

Table 8 shows the association between glucose variables during 72 h and neurological outcomes. Maximum glucose (aOR, 1.004; 95% CIs, 1.001–1.007), mean glucose (aOR, 1.009; 95% CIs, 1.002–1.016), andSD (aOR, 1.019; 95% CIs, 1.005–1.032) of glucose were independently associated with poor neurological outcomes.

Adjusted for age, first monitored rhythm, etiology of cardiac arrest, time from collapse to ROSC, PaO$_2$ after ROSC, PaCO$_2$ after ROSC, and pre-targeted temperature management shock

**Table 6. The association between insulin administration method and poor neurological outcomes.**

| Variables | Adjusted odds ratio (95% confidence interval) | p |
|---|---|---|
| Age, years | 1.037 (1.015–1.059) | 0.001 |
| Diabetes mellitus | 0.792 (0.427–1.471) | 0.460 |
| Renal disease | 1.469 (0.461–4.680) | 0.515 |
| Witnessed | 0.776 (0.378–1.590) | 0.488 |
| Shockable rhythm | 0.182 (0.099–0.336) | < 0.001 |
| Cardiac etiology | 0.203 (0.083–0.496) | < 0.001 |
| Time to ROSC, min | 1.063 (1.042–1.083) | < 0.001 |
| $PaO_2$, mmHg | 1.003 (1.000–1.005) | 0.049 |
| $PaCO_2$, mmHg | 1.014 (1.001–1.028) | 0.036 |
| Pre-TTM shock | 1.632 (0.933–2.854) | 0.086 |
| Target temperature | 1.240 (0.533–2.882) | 0.617 |
| 33˚C–34˚C | Reference | |
| 35˚C–36˚C | | |
| Insulin administration method | | |
| CII | Reference | |
| SQI | 0.660 (0.335–1.301) | 0.230 |
| IBI | 1.757 (0.867–3.560) | 0.118 |

ROSC, return of spontaneous circulation; TTM, targeted temperature management; CII, continuous intravenous insulin; SQI, subcutaneous insulin; IBI, intravenous bolus insulin

**Table 7. Glucose characteristics during 72 h after cardiac arrest stratified by neurological outcomes.**

| Characteristics | Good (n = 111) | Poor (n = 438) | p |
|---|---|---|---|
| Maximum, mg/dL | 308 (250–366) | 343 (284–410) | <0.001 |
| Mean, mg/dL | 164 (145–189) | 179 (154–216) | <0.001 |
| SD | 41.1 (29.9–52.9) | 48.2 (35.9–67.0) | <0.001 |
| Minimum, mg/dL | 98 (89–111) | 100 (82–116) | 0.590 |
| Moderate hypoglycemia | 6 (5.4) | 52 (11.9) | 0.048 |
| Severe hypoglycemia | 4 (3.6) | 13 (3.0) | 0.730 |

SD, standard deviation

**Table 8. The association between glucose variables during 72 h and neurological outcomes.**

| Characteristics | Adjusted odds ratio (95% confidence interval) | p |
|---|---|---|
| Maximum | 1.004 (1.001–1.007) | 0.004 |
| Mean | 1.009 (1.002–1.016) | 0.013 |
| SD | 1.019 (1.005–1.032) | 0.006 |
| Minimum | 1.002 (0.993–1.011) | 0.688 |
| Moderate hypoglycemia | 1.331 (0.457–3.878) | 0.600 |
| Severe hypoglycemia | 0.856 (0.176–4.162) | 0.847 |

SD, standard deviation

## Discussion

This retrospective analysis found that the insulin administration method had no association with the achievement of mean glucose ≤ 180 mg/dL and the 6-month neurological outcome. Nevertheless, the CII group had the lowest mean glucose levels, and the IBI group had the highest SD of glucose levels. The CII and IBI groups had lower minimum glucose levels than the SQI group. The IBI group had a higher incidence of severe hypoglycemia. Higher maximum glucose, mean glucose, and SD of glucose were independently associated with poor neurological outcomes.

CII is the preferred route and delivery method of insulin in critically ill patients with hyperglycemic crises, such as perioperative care of cardiac surgery, cardiogenic shock, myocardial infarction, and acute ischemic stroke [16]. Such situations may require a rapid change in the insulin level or be associated with poor perfusion of subcutaneous tissue; for these reasons, guidelines recommend CII in critically ill patients instead of SQI [17]. A study comparing CII to SQI in patients with diabetes undergoing coronary artery bypass grafting demonstrated that CII was associated with lower postoperative blood glucose and reduced mortality [9]. Nevertheless, SQI is still used to control blood glucose in critically ill patients [18]. Likewise, IBI has a comparable effect on blood glucose to CII without any adverse effect [19]. Although CII was used the most, SQI and IBI were also used in about half of the patients in the present study. We found that the SQI and IBI groups had higher mean blood glucose levels during the 72 h after TTM initiation than the CII group. Higher mean blood glucose level was associated with the 6-month poor neurological outcomes in the present study. However, SQI or IBI had no association with the achievement of mean glucose ≤ 180 mg/dL and the 6-month neurological outcomes compared to CII. It might be postulated that SQI or IBI is not inferior to CII regarding controlling hyperglycemia, considering the independent association between high mean glucose and poor neurological outcome.

Regardless of the relationship of the insulin administration method with clinical outcomes, regarding glycemic control, CII seems to be the best measure among the three insulin administration methods. Hyperglycemia early after ROSC, at admission, or during 36 h after admission in survivors of cardiac arrest who underwent therapeutic hypothermia or TTM was associated with poor neurological outcomes [2, 4, 5]. The blood glucose level after ROSC was not associated with the neurological outcomes in the present study. We postulate that because we excluded the patients without hyperglycemia within 24 h after ROSC, it might affect the insignificant relationship between glucose after ROSC and neurological outcomes; this is because the patients with no hyperglycemia after ROSC would be injured less than those with hyperglycemia due to ischemic insult. However, consistent with the previous study reports, high maximum and mean glucose levels during the 72 h following initiation of TTM were associated with poor neurological outcomes in the present study. Our results strengthen the importance of glucose control for a more extended period following ROSC.

Blood glucose level and glucose variability have been assessed in comatose cardiac arrest survivors, and high glucose variability is associated with poor neurological outcomes [2, 3, 5]. We calculated the SD of glucose for 72 h after the initiation of TTM to determine glucose variability. We also found that high SD of glucose was associated with poor neurological outcomes in the present study. Although there was no association between the insulin administration method and neurological outcomes, lower mean glucose levels and lower SD of glucose associated with favorable neurological outcomes emphasize that the CII may be the best measure to control glycemic status in cardiac arrest survivors.

A landmark trial regarding glucose control in critically ill patients revealed that intensive glucose control (81 to 110 mg/dL) increased mortality in the intensive care unit, and the high

mortality in the intensive glucose control group was associated with iatrogenic hypoglycemia [6, 20]. Therefore, avoiding hypoglycemia is as crucial as avoiding hyperglycemia when controlling blood glucose. Although we found no association between neurological outcomes and moderate or severe hypoglycemia, the IBI and CII groups had frequent severe hypoglycemia compared to the SQI group in the present study. Frequent blood glucose monitoring is required more in the IBI and CII groups than in the SQI group to avoid hypoglycemia. Conventional glucose control (target ≤ 180 mg/dL) resulted in total 16.3% of hypoglycemia (moderate hypoglycemia, 15.8%; severe hypoglycemia, 0.5%) in a randomized control trial [6]. We reported 13.7% of hypoglycemia (moderate hypoglycemia, 10.6%; severe hypoglycemia, 3.1%). In the randomized control group, patients in the conventional glucose control group had a history of diabetes mellitus in about 20%, whereas we reported that 35.5% of the patients had a history of diabetes mellitus [6]. Diabetes mellitus does not seem to play a role in hypoglycemia during insulin treatment.

This study has several limitations. Although we used the data from the prospective multicenter registry, we can only demonstrate an association because this is a retrospective analysis. Of the total registry, about 60% of the patients were excluded, which might have caused selection bias. The registry is configured to record blood glucose every 4 h during TTM. Therefore, glucose characteristics might be different from the actual values. Although diabetes mellitus was not associated with neurological outcomes, pre-existing diabetes mellitus differed among the insulin treatment groups. Glycated hemoglobin has been reported to be associated with neurological outcomes in comatose cardiac arrest survivors [21], which means that glucose control status before cardiac arrest rather than the history of diabetes mellitus might contribute to neurological outcomes. However, we could not analyze the data according to glycated hemoglobin level due to the limitation of data. Future studies need to address the interaction of the insulin administration method with glycemic control status. The target temperature differed among the insulin treatment groups because the target temperature and the insulin administration method depended on the post-cardiac arrest care protocol at each hospital rather than the attending physician. However, the target temperature was not associated with neurological outcomes in the present study. The targeted temperature management trial also showed no difference in blood glucose and glucose variability between the target temperature groups [2]. Even between CII and IBI, the detailed administration method is different. The total insulin dosage may vary depending on the insulin administration method. Due to data limitations, it was impossible to compare the total amount of insulin administered in the present study.

## Conclusions

The insulin administration method was not associated with the achievement of mean glucose ≤ 180 mg/dL and the 6-month neurological outcomes in OHCA survivors who underwent TTM. The CII method had lower mean glucose and lower SD of glucose rather than the IBI and SQI methods.

## Supporting information

**S1 File. Insulin administration protocols.** The participating hospitals in Korean Hypothermia Network have their insulin administration protocol. The protocols are a bit different from each other.
(PDF)

**S1 Data. Raw data.**
(XLSX)

## Acknowledgments

The following investigators participated in the Korean Hypothermia Network. Chair: Kyung Woon Jeung (Chonnam National University Hospital, E-mail: neoneti@hanmail.net). Principal investigators of each hospital: Kyu Nam Park (The Catholic University of Korea, Seoul St. Mary's Hospital); Minjung Kathy Chae (Ajou University Medical Center); Won Young Kim (Asan Medical Center); Byung Kook Lee (Chonnam National University Hospital); Dong Hoon Lee (Chung-Ang University Hospital); Tae Chang Jang (Daegu Catholic University Medical Center); Jae Hoon Lee (Dong-A University Hospital); Yoon Hee Choi (Ewha Womans University Mokdong Hospital); Je Sung You (Gangnam Severance Hospital); Young Hwan Lee (Hallym University Sacred Heart Hospital); In Soo Cho (Hanil General Hospital); Su Jin Kim (Korea University Anam Hospital); Jong-Seok Lee (Kyung Hee University Medical Center); Yong Hwan Kim (Samsung Changwon Hospital); Min Seob Sim (Samsung Medical Center); Jonghwan Shin (Seoul Metropolitan Government Seoul National University Boramae Medical Center); Yoo Seok Park (Severance Hospital); Hyung Jun Moon (Soonchunhyang University Hospital Cheonan); Won Jung Jeong (The Catholic University of Korea, St. Vincent's Hospital); Joo Suk Oh (The Catholic University of Korea, Uijeongbu St. Mary's Hospital); Seung Pill Choi (The Catholic University of Korea, Yeouido St. Mary's Hospital); Kyoung-Chul Cha (Wonju Severance Christian Hospital).

## Author Contributions

**Conceptualization:** Byung Kook Lee.

**Data curation:** Yong Soo Cho, Yong Hun Jung, Chun Song Youn, Soo Hyun Kim.

**Formal analysis:** Dong Hun Lee, Byung Kook Lee.

**Funding acquisition:** Byung Kook Lee.

**Investigation:** Dong Hun Lee, Byung Kook Lee, Hyoung Youn Lee, Kyung Woon Jeung.

**Methodology:** Soo Hyun Kim.

**Project administration:** Chun Song Youn.

**Supervision:** Kyung Woon Jeung.

**Writing – original draft:** Dong Hun Lee, Byung Kook Lee.

**Writing – review & editing:** Yong Soo Cho, Yong Hun Jung, Hyoung Youn Lee, Kyung Woon Jeung, Chun Song Youn, Soo Hyun Kim.

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
