## [Decision Letter · Decision Letter 0]

16 Aug 2022

PONE-D-22-17886Association between insulin administration method and six-month neurological outcome in survivors of out-of-hospital cardiac arrest who underwent targeted temperature managementPLOS ONE

Dear Dr. Lee,

Thank you for submitting your manuscript to PLOS ONE. After careful consideration, we feel that it has merit but does not fully meet PLOS ONE’s publication criteria as it currently stands. Therefore, we invite you to submit a revised version of the manuscript that addresses the points raised during the review process.

ACADEMIC EDITOR: Thank you very much for having submitted this paper. As you can see in the comments one of the Reviewer has raised many concerns about the present version of the paper.  I would give you the opportunity do address the comments. 

We look forward to receiving your revised manuscript.

Kind regards,

Simone Savastano

Academic Editor

PLOS ONE

Journal Requirements:

"This study was supported by a grant from National Research Foundation of Korea (NRF-2021R1A2C1005800)."

Additional Editor Comments:

Thank you very much for having submitted this paper. As you can see in the comments one of the Reviewer has raised many concerns about the present version of the paper. I would give you the opportunity do address the comments.

Reviewers' comments:

Reviewer's Responses to Questions

**Comments to the Author**

1. Is the manuscript technically sound, and do the data support the conclusions?

Reviewer #1: Yes

Reviewer #2: Partly

2. Has the statistical analysis been performed appropriately and rigorously? 

Reviewer #1: Yes

Reviewer #2: No

3. Have the authors made all data underlying the findings in their manuscript fully available?

Reviewer #1: No

Reviewer #2: Yes

4. Is the manuscript presented in an intelligible fashion and written in standard English?

Reviewer #1: Yes

Reviewer #2: Yes

5. Review Comments to the Author

Reviewer #1: This is a retrospective study (although the data were collected prospectively in a multicenter registry) comparing different methods of glucose administration (subcutaneous -SQI, intravenous bolus infusion -IBI, continuous intravenous infusion -CII) in patients admitted for OHCA (out of hospital cardiac arrest) undergoing targeted temperature management.

Insulin administration method was not associated with the 6-month neurological outcomes, however the continuous infusion was associated with lower mean glucose and lower standard deviation of glucose.

Moreover (even if that was not the endpoint of the study), it adds on the evidence that glucose variables (mean value, maximum value, SD, episodes of moderate hypoglycemia) are associated with neurological outcome after OHCA.

It is also interesting to note that SQI and IBI method were associated with a mean glucose level (respectively 181 and 184 mg/dl) higher than the cutoff of <180 mg/dl recommended in the guidelines, being the CII method the only one with a mean glucose level in target (168 mg/dl) according to the guidelines.

Finally, although there was no association between the insulin administration method and neurological outcomes, lower mean glucose levels and lower SD of glucose were associated with favorable neurological outcomes and CII was associated with ower mean glucose levels and lower SD of glucose, suggesting that the CII may be the best measure to control glycemic status in cardiac arrest survivors.

Some issues come to mind:

- It would be interesting to know if there were statistical differences in terms of total unit of insulin administered between the three methods.

- At line 261 “A landmark trial regarding glucose control in critically ill patients revealed that intensive glucose control (< 180 mg/dL) increased mortality in the intensive care unit” is probably a typo since, according to the reference (6), the intensive glucose control in the trial was between 81 and 10mg/dl.

Other than this minor issues, the article seems fit for publication

Reviewer #2: Review PONE-D-22-17886

In this article authors retrospectively investigate the association between insulin administration method and neurological outcomes in out-of-hospital cardiac arrest (OHCA) survivors; data were extracted from a multicentre registry of patients with OHCA who underwent targeted temperature management (TTM). Authors ambitiously tried to use Cerebral Performance Category Scale as primary endpoint, evaluating its association with insulin administration method. Insulin regulating mechanism is known to be altered in patient with cardiac arrest.

However, the physiopathological mechanism according to which it has been hypothesized that insulin administration method would be associated with neurological outcome is not clearly described: maybe one way of administration (CII) should better achieve the optimal glucose target? In this case, first step analysis should address this physiopathological mechanism; the association between insulin administration method and neurological outcomes in OHCA should be investigated secondarily.

Major issues

1. The three different groups according to way of insulin administration (SQUI, IBI and CII) are characterized by significantly different glucose values, prevalence of diabetes mellitus and other parameters which are described to influence neurological outcome (witnesses to collapse, cardiac aetiology, time from collapse to ROSC). Based on above findings, the groups can't be considered homogeneous.

2. Why patient with adequate glucose control with insulin therapy within 24 hours from ROSC were not included?

6. PLOS authors have the option to publish the peer review history of their article (what does this mean?). If published, this will include your full peer review and any attached files.

Reviewer #1: No

Reviewer #2: No

---

## [Author Response · Author response to Decision Letter 0]

19 Sep 2022

Dear editor, we revised the manuscript according to the requirement of the journal.

Editor

Answer) We revised the manuscript according to the PLOS One style requirements. 

"This study was supported by a grant from National Research Foundation of Korea (NRF-2021R1A2C1005800)."

Answer) We added the sentence as the editor advised in the Funding Statement. 

3. In your Data Availability statement, you have not specified where the minimal data set underlying the results described in your manuscript can be found. PLOS defines a study's minimal data set as the underlying data used to reach the conclusions drawn in the manuscript and any additional data required to replicate the reported study findings in their entirety. All PLOS journals require that the minimal data set be made fully available. We will update your Data Availability statement to reflect the information you provide in your cover letter.

Answer) we added the minimal data set as a Supplementary file.

Dear reviewers, we appreciate your kind review. They were very helpful in improving our manuscript. After due consideration, the manuscript was revised as described below. 

Response to Reviewers

Reviewer #1: 

1. It would be interesting to know if there were statistical differences in terms of total unit of insulin administered between the three methods.

Answer) As you indicated, it would be interesting to know whether the amount of insulin administered varies depending on the insulin administration methods. We also thought that the topic would be very relevant. However, the registry was built to input insulin dose data optionally. Therefore, calculating insulin dose using our data is inappropriate because the input of insulin dose information was not completed, and insulin administered more frequently than every 4 hrs is not entered. 

We added the following sentence in the limitation

“The total insulin dosage may vary depending on the insulin administration method. Due to data limitations, it was impossible to compare the total amount of insulin administered in the present study.”

2. At line 261 “A landmark trial regarding glucose control in critically ill patients revealed that intensive glucose control (< 180 mg/dL) increased mortality in the intensive care unit” is probably a typo since, according to the reference (6), the intensive glucose control in the trial was between 81 and 10mg/dl.

Answer) Thank you for finding the typo. We revised as “intensive glucose control (81 to 110 mg/dL)

Reviewer #2: Review PONE-D-22-17886

In this article authors retrospectively investigate the association between insulin administration method and neurological outcomes in out-of-hospital cardiac arrest (OHCA) survivors; data were extracted from a multicentre registry of patients with OHCA who underwent targeted temperature management (TTM). Authors ambitiously tried to use Cerebral Performance Category Scale as primary endpoint, evaluating its association with insulin administration method. Insulin regulating mechanism is known to be altered in patient with cardiac arrest. However, the physiopathological mechanism according to which it has been hypothesized that insulin administration method would be associated with neurological outcome is not clearly described: maybe one way of administration (CII) should better achieve the optimal glucose target? In this case, first step analysis should address this physiopathological mechanism; the association between insulin administration method and neurological outcomes in OHCA should be investigated secondarily.

Major issues

1. The three different groups according to way of insulin administration (SQI, IBI and CII) are characterized by significantly different glucose values, prevalence of diabetes mellitus and other parameters which are described to influence neurological outcome (witnesses to collapse, cardiac aetiology, time from collapse to ROSC). Based on above findings, the groups can't be considered homogeneous.

Answer) Considering the pathophysiologic mechanism, as suggested by the reviewer, the primary outcome was mean glucose ≤ 180 mg/dL. We added multivariate regression analysis, which investigated the association between the insulin administration method and mean glucose ≤ 180 mg/dL after adjusting the covariates. 

To control the heterogenous characteristics among three insulin administration methods, variables that are different in the univariate analysis (comparison among insulin administration methods) and variables that are independently associated with mean glucose ≤ 180 mg/dL and neurological outcomes were also included in the multivariable logistic regression analysis. Therefore, we added two tables and results and reordered the tables and results. 

Accordingly, we revised the manuscript. 

The following sentence in the Introduction 

“To address this question, we hypothesized that the insulin administration method would be associated with neurological outcomes and changes in blood glucose.”

Was revised as following

“To address this question, we hypothesized that the insulin administration method would be related to blood glucose levels and thus to neurological outcomes.”

The following sentences in the Method

“The primary outcome was neurological outcomes assessed using CPC 6 months after ROSC and defined as good (CPC 1 or 2) or poor (CPC 3–5). The secondary outcomes were maximum glucose, mean glucose, a standard deviation (SD) of glucose, minimum glucose, moderate hypoglycemia, and severe hypoglycemia.” 

Was revised as following

“The primary outcome was the achievement of mean blood glucose < 180 mg/dL. The secondary outcomes were neurological outcomes assessed using CPC 6 months after ROSC, maximum glucose, mean glucose, a standard deviation (SD) of glucose, minimum glucose, moderate hypoglycemia, and severe hypoglycemia. The neurological outcomes were defined as good (CPC 1 or 2) or poor (CPC 3–5).”

The following sentence in the Statistical analysis

“We conducted a logistic regression analysis to investigate the association between insulin administration methods and neurological outcomes.”

Was revised as following

“We conducted logistic regression analyses to investigate the association between the insulin administration method and the achievement of mean blood glucose ≤ 180 mg/dL and the association between insulin administration methods and neurological outcomes.”

The following sentences in the Statistical analysis 

“All variables with a p-value < 0.2 in univariate analyses were included in the multivariate logistic regression analysis to find the covariates. We performed the Hosmer–Lemeshow test to test the goodness-of-fit of the logistic model. We selected age, first monitored rhythm, etiology of cardiac arrest, time from collapse to ROSC, PaO2 after ROSC, PaCO2 after ROSC, and pre-TTM shock as covariates those had p-value < 0.05 in the multivariate logistic regression analysis. We conducted the logistic regression analysis with insulin administration methods and covariates.”

Was revised as following

“We selected the variables with a p-value < 0.05 in comparisons among insulin administration methods as covariates for the association between insulin administration methods and mean blood glucose ≤ 180 mg/dL. Additionally, we selected the covariates after performing the multivariable logistic regression analysis with the variables with a p-value < 0.2 in comparisons between groups of mean glucose ≤ 180 mg/dL and > 180 mg/dL. We also selected the covariates for the association between insulin administration methods and neurological outcomes through the multivariable logistic regression analysis with variables with a p-value <0.2 in comparisons between neurological outcome groups. We selected the variables with a p-value < 0.05 in the multivariate logistic regression analyses as final covariates. We performed the Hosmer–Lemeshow test to test the goodness-of-fit of the logistic model.”

We reordered the Tables and added table 3 and table 4, which compares characteristics between mean glucose ≤ 180 mg/dL and mean glucose > 180 mg/dL and multivariable regression analysis between insulin administration methods and mean glucose ≤ 180 mg/dL

Table 2 � table 1

Table 3 � table 2

Table 1 � table 5

Table 4 � table 6

Table 5 � table 7

Table 6 � table 8

Glucose after ROSC among three insulin administration methods was moved to Table 1.

Mean glucose ≤ 180 mg/dL was added in the table 2 and revised the following sentence in the result. 

The following sentence in the Results

“Mean glucose, SD of glucose, and minimum glucose were different among the three groups.”

Was revised as following

“Achievement of mean glucose ≤ 180 mg/dL, Mean glucose, SD of glucose, and minimum glucose differed among the three groups.”

The following tables (Table 3 and Table 4) and paragraphs were added. 

“Association between insulin administration method and mean glucose under 180 mg/dL

 Table 3 shows the demographic and clinical characteristics stratified by mean glucose of 180 mg/dL. A total of 296 (53.9%) achieved mean glucose ≤ 180 mg/dL. Patients who had mean glucose ≤ 180 mg/dL had lower BMI (23.0 kg/m2 [20.7–25.4]) vs. 23.9 kg/m2 [21.3–26.1]) than those had mean glucose > 180 mg/dL. Patients with mean glucose ≤ 180 mg/dL had fewer pre-existing illnesses (hypertension and diabetes mellitus). Patient who had mean glucose ≤ 180 mg/dL fewer witnesses of collapse (208/296 vs. 154/253), lower epinephrine dose (2 mg [0–4] vs. 2 mg [1–5]), lower serum lactate (9.3 mg/dL [6.0–12.6] vs. 11.3 mg/dL [8.0–14.0]), higher PaO2 (117.5 mmHg [76.0–221.0] vs. 100.0 mmHg [71.0–160.0]), lower pre-TTM shock (155/296 vs. 176/253), lower SOFA scores (11 [9–13] vs. 12 [10–14]), and a higher proportion of target temperature of 33℃–34℃ (255/296 vs. 192/253).” 

“Multivariable logistic regression analysis found hypertension (aOR, 0.609; 95% CIs, 0.399–0.930), diabetes mellitus (aOR, 0.273; 95% CIs, 0.175–0.425), witnessed (aOR, 1.992; 95% CIs, 1.289–3.078), PaO2 (aOR, 1.002; 95% CIs, 1.000–1.004), pre-TTM shock (aOR, 0.541; 95% CIs, 0.355–0.826), target temperature of 35–36℃ (aOR, 0.500; 95% CIs, 0.297–0.842), glucose after ROSC (aOR, 0.995; 95% CIs, 0.993–0.997) as covariates. The insulin administration method was not associated with the achievement of mean glucose ≤ 180 mg/dL after adjusting covariates (Table 4).”

Table 4 was revised as Table 6 as following

The sentences in the Discussion

“This retrospective analysis found that the insulin administration method had no association with the 6-month neurological outcome. The CII group had the lowest mean glucose levels and the IBI group had the highest SD of glucose levels.”

Was revised as following

“This retrospective analysis found that the insulin administration method had no association with the achievement of mean glucose ≤ 180 mg/dL and the 6-month neurological outcome. Nevertheless, the CII group had the lowest mean glucose levels and the IBI group had the highest SD of glucose levels.” 

The sentence in the Discussion

“However, SQI or IBI had no association with the 6-month neurological outcomes compared to CII. This discrepancy between blood glucose variables and clinical outcomes according to the insulin administration method warrants further clinical trials.”

Was revised as following

“However, SQI or IBI had no association with the achievement of mean glucose ≤ 180 mg/dL and the 6-month neurological outcomes compared to CII. It might be postulated that SQI or IBI is not inferior to CII regarding controlling hyperglycemia, considering the independent association between high mean glucose and poor neurological outcome.” 

The sentence in the Conclusion

“The insulin administration method was not associated with the 6-month neurological outcomes in OHCA survivors who underwent TTM. The CII method was associated with lower mean glucose and lower SD of glucose rather than the IBI and SQI methods.”

Was revised as following

“The insulin administration method was not associated with the achievement of mean glucose ≤ 180 mg/dL and the 6-month neurological outcomes in OHCA survivors who underwent TTM. The CII method had lower mean glucose and lower SD of glucose rather than the IBI and SQI methods.” 

Abstract was revised as following

“We investigated the association of insulin administration method with the achievement of mean glucose ≤ 180 mg/dL and neurological outcomes in out-of-hospital cardiac arrest (OHCA) survivors who had hyperglycemia after the return of spontaneous circulation. From a multicenter prospective registry, we extracted the data of adult OHCA survivors who underwent targeted temperature management (TTM) between 2015 and 2018. Blood glucose levels every 4 h after initiating TTM were obtained for 72 h. We divided insulin administration methods into three categories: subcutaneous (SQI), intravenous bolus (IBI), and continuous intravenous (CII). We calculated the mean glucose and standard deviation (SD) of glucose. The primary outcome was the achievement of mean glucose ≤ 180 mg/dL. The secondary outcomes were the 6-month neurological outcome based on the Cerebral Performance Category (CPC) scale (good, CPC 1–2; poor, CPC 3–5), mean glucose, and SD of glucose. Of the 549 patients, 296 (53.9%) achieved mean glucose ≤ 180 mg/dL, and 438 (79.8%) had poor neurological outcomes, 134 (24.4%), 132 (24.0), and 283 (51.5%) were in the SQI, IBI, and CII groups, respectively. The SQI (adjusted odds ratio [aOR], 0.848; 95% confidence intervals [CIs], 0.493–1.461) and IBI (aOR, 0.673; 95% CIs, 0.415–1.091) groups were not associated with mean glucose ≤ 180 mg/dL and the SQI (aOR, 0.660; 95% CIs, 0.335–1.301) and IBI (aOR, 1.757; 95% CIs, 0.867–3.560) groups were not associated with poor neurological outcomes compared to the CII group. The CII (168 mg/dL [147–202]) group had the lowest mean glucose than the SQI (181 mg/dL [156–218]) and IBI (184 mg/dL [162–216]) groups. The CII (45.0[33.9–63.5]) group had a lower SD of glucose than the IBI (50.8 [39.1–72.0]) group. The insulin administration method was not associated with achieving mean glucose ≤ 180 mg/dL and 6-month neurological outcomes.” 

2. Why patient with adequate glucose control with insulin therapy within 24 hours from ROSC were not included?

Answer) We excluded the patient who had no hyperglycemia (>180 mg/dL) within 24 h after ROSC. In other words, we excluded the patients who did not require insulin administration to control the blood glucose. However, the patients who were given insulin to control blood glucose within 24 h after ROSC were included irrespective of adequate glucose control.

---

## [Decision Letter · Decision Letter 1]

14 Dec 2022

Association between insulin administration method and six-month neurological outcome in survivors of out-of-hospital cardiac arrest who underwent targeted temperature management

PONE-D-22-17886R1

Dear Dr. Lee,

We’re pleased to inform you that your manuscript has been judged scientifically suitable for publication and will be formally accepted for publication once it meets all outstanding technical requirements.

Kind regards,

Simone Savastano

Academic Editor

PLOS ONE

Additional Editor Comments (optional):

Thank you very much for having submitted this paper and for having addressed all the reviewers' comments. I congratulate you on work work

Reviewers' comments:

Reviewer's Responses to Questions

**Comments to the Author**

1. If the authors have adequately addressed your comments raised in a previous round of review and you feel that this manuscript is now acceptable for publication, you may indicate that here to bypass the “Comments to the Author” section, enter your conflict of interest statement in the “Confidential to Editor” section, and submit your "Accept" recommendation.

Reviewer #1: All comments have been addressed

Reviewer #2: All comments have been addressed

2. Is the manuscript technically sound, and do the data support the conclusions?

Reviewer #1: Yes

Reviewer #2: Partly

3. Has the statistical analysis been performed appropriately and rigorously? 

Reviewer #1: Yes

Reviewer #2: Yes

4. Have the authors made all data underlying the findings in their manuscript fully available?

Reviewer #1: (No Response)

Reviewer #2: Yes

5. Is the manuscript presented in an intelligible fashion and written in standard English?

Reviewer #1: Yes

Reviewer #2: Yes

6. Review Comments to the Author

Reviewer #1: All issues were addressed and typos corrected, overall the article seems now fit for publication.

Reviewer #2: All comments have been adressed. However, since primary purpose and endpoint have been changed, the manuscript title must be modified.

7. PLOS authors have the option to publish the peer review history of their article (what does this mean?). If published, this will include your full peer review and any attached files.

Reviewer #1: No

Reviewer #2: No

---

## [Editor Report · Acceptance letter]

21 Dec 2022

PONE-D-22-17886R1 

Association between insulin administration method and six-month neurological outcome in survivors of out-of-hospital cardiac arrest who underwent targeted temperature management 

Dear Dr. Lee:

I'm pleased to inform you that your manuscript has been deemed suitable for publication in PLOS ONE. Congratulations! Your manuscript is now with our production department. 

Kind regards, 

on behalf of

Dr. Simone Savastano 

Academic Editor

PLOS ONE